# OntoTune: Ontology-Driven Self-training for Aligning Large Language Models

## ABSTRACT

Existing domain-specific Large Language Models (LLMs) are typically developed by fine-tuning general-purposed LLMs with large-scale domain-specific corpora. However, training on large-scale corpora often fails to effectively organize domain knowledge of LLMs, leading to fragmented understanding. Inspired by how humans connect concepts and organize knowledge through mind maps, we aim to emulate this approach by using **ontology** with hierarchical conceptual knowledge to reorganize LLM's domain knowledge. From this perspective, we propose an ontology-driven self-training framework called **OntoTune**, which aims to align LLMs with ontology through in-context learning, enabling the generation of responses guided by the ontology. We leverage in-context learning to identify whether the LLM has acquired the specific concept's ontology knowledge, and select the entries not yet mastered by LLM as the training set to further align the LLM with ontology. Compare to existing domain LLMs based on newly collected large-scale domain-specific corpora, our OntoTune, which relies on the existing, long-term developed ontology and LLM itself, significantly reduces data maintenance costs and offers improved generalization ability. We conduct our study in the medical domain to evaluate the effectiveness of OntoTune, utilizing a standardized medical ontology, SNOMED CT as our ontology source. Experimental results demonstrate that OntoTune achieves state-of-the-art performance in both in-ontology task hypernym discovery and out-of-ontology task medical domain QA. Moreover, compared to the latest direct ontology injection method TaxoLLaMA, our OntoTune better preserves original knowledge of LLM.

## CCS CONCEPTS

• **Computing methodologies** → *Natural language processing*; *Semantic networks*.

## KEYWORDS

Large Language Model, Self-training, Align with Ontology

**ACM Reference Format:**
Anonymous Author(s). 2018. OntoTune: Ontology-Driven Self-training for Aligning Large Language Models. In *Proceedings of Make sure to enter the correct conference title from your rights confirmation emai (Conference acronym 'XX)*. ACM, New York, NY, USA, 17 pages. https://doi.org/XXXXXXX.XXXXXXX

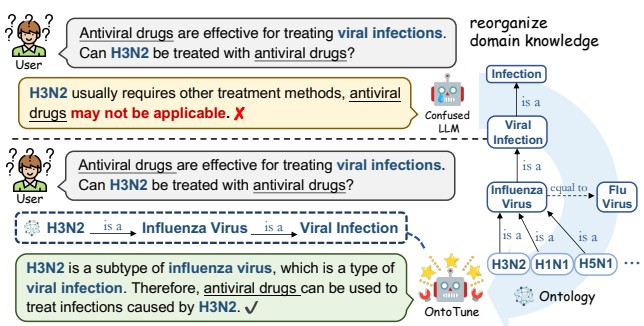

**Figure 1: A simple example illustrates how hierarchical structure knowledge in the ontology guide responses.**

## 1 INTRODUCTION

Large Language Models (LLMs), such as GPT-4 [41] and LLaMA [15], have achieved remarkable success in the field of natural language processing [51], demonstrating advanced performance across various domains and tasks. To further enhance the capabilities of LLMs in specific domain, such as medical, financial, and science, the research and industry community have begun to focus on developing domain-specific LLMs [2, 7, 31].

Existing methods usually develop domain-specific LLMs by further training general-purposed LLMs on domain- specific corpora, such as BloombergGPT [55], BioMistral [31] and LawGPT [67]. Previous researches [16, 45] indicate that LLMs have already acquired most domain knowledge during the comprehensive pre-training phase, and need to reorganize and align knowledge with domain-specific requirements during the post-training phase. However, adapting LLMs to specific domains presents significant challenges [32, 66]. On the one hand, the scarcity of domain-specific corpora and constraints imposed by data privacy present significant hurdles in the continuous collection of high-quality domain corpora for continual pre-training or supervised fine-tuning, demanding substantial investment in time and resources. On the other hand, existing researches [10, 14] reveal that directly fine-tuning LLMs with fragmented raw domain corpora struggles to effectively organize domain knowledge and can even impair prompting capabilities of LLMs. **So can we find a more efficient alternative to reorganize domain knowledge in large language models without relying on large-scale domain-specific corpora?**

*Inspired by how humans use mind maps which visually represent concepts and their relationships, to systematically organize and review knowledge, we aim to use domain-specific mind maps to reorganize LLM's domain knowledge.* Naturally, we associate these mind map with widely established, rigorously constructed **ontologies** [56], which fully display the relationships and hierarchical structures between domain concepts as the ideal domain-specific mind maps. As shown in Figure 1, the ontology structure primarily consists of

hypernym and synonym relationships between concepts, and have been widely applied in scenarios such as information retrieval [58] and knowledge reasoning [26, 62]. Common domain ontologies include SNOMED CT [46] in the biomedical field, WordNet [37] in the lexical field and GeoNames[1] in the geographical field. Figure 1 illustrates an example of a medical ontology guided response, where the LLM links concepts through the hierarchical structure knowledge in the ontology. Meanwhile, we suppose that compared to collecting new large-scale domain corpora, utilizing existing, long-term developed ontologies can reduce data maintenance costs and offer better generalization. From this perspective, we propose an ontology-driven self-training fine-tuning framework **OntoTune**, which aims to align LLMs with domain ontology through in-context learning[2] and generate responses guided by the ontology. Onto-Tune's workflow consists of three main steps: **(1) Instruction Text Generation.** We utilize three ontology-aware concept-level instructions which focus respectively on diversity, conceptuality, and professionalism to generate outputs. Then we incorporate the corresponding ontology knowledge to the input and let seed model rethink to obtain better outputs through in-context learning. **(2) Inconsistency Text Selection.** If there is significant inconsistency between the corpora obtained with and without ontology knowledge, we consider that the seed model has not effectively grasped this concept's ontology structure to guide its output and select entries that exhibit significant inconsistency as the training set. **(3) LLM Fine-tuning.** Based on the training set, we perform self-training on the seed model, resulting in aligned domain LLMs.

We conduct our study in the medical field, using the high-quality medical ontology SNOMED CT [46] as the source ontology. To evaluate the effectiveness of OntoTune, we compare it not only with customized models for specific tasks but also with existing domain LLM trained on large-scale corpora and the direct ontology injection method TaxoLLaMA* [38] implemented on the same LLM called seed model. Results show that we have achieved state-of-the-art performance in in-ontology task hypernym discovery and out-of-ontology task domain QA, demonstrating that OntoTune can effectively improve the performance of domain-specific tasks. Moreover, OntoTune significantly preserves the knowledge and safety of the seed model compared to existing domain-specific LLMs and TaxoLLaMA. Our contributions can be summarized as follows:

- We highlight the limitations of developing domain LLMs based on large-scale domain corpora, and we are the first to utilize small-scale ontology to reorganize the domain knowledge of LLMs.

- We propose a novel ontology-driven self-training method OntoTune, which aligns LLMs with ontologies through in-context learning, thereby guiding LLMs to generate responses under domain ontology knowledge.

- Compared to exsiting domain LLM based on large-scale raw domain corpora and the direct injection method TaxoLLaMA, our OntoTune achieves state-of-the-art performance in the in-ontology task hypernym discovery and out-of-ontology task domain QA, and significantly preserves the knowledge capabilities and safety of the seed model.

---

[1]https://www.geonames.org/
[2]https://openai.com/index/learning-to-reason-with-llms/

## 2 RELATED WORKS

### 2.1 Domain-specific LLMs

Existing domain-specific large language models (LLMs) can be categorized into two groups: (1) those models trained from scratch using domain-specific corpora, such as BioGPT [35] and GatorTron [59], and (2) those [9, 31, 55, 63] that employ continual training on general-purposed models. Benefiting from its ability to leverage the extensive and diverse data of the seed models, as well as more efficient training processes, the latter approach has gradually become mainstream. Current domain-specific LLMs like BioMistral [31], BloombergGPT [55] and LawGPT [67] are developed by training a seed model with a large-scale raw domain-specific corpora, demonstrating impressive performance on domain tasks. To be specific, the medical model PMC-LLaMA [54] is fine-tuned with LoRA [23] on LLaMA using 4.8 million biomedical papers. LawGPT [67] continues training on 500k legal documents. And BloombergGPT [55] is fine-tuned on a 708 billion tokens financial corpora. These models typically rely on large amounts of training data to adapt to their respective domains. However, this fragmented knowledge from the raw corpora is merely injected into the seed model without being systematically organized and recent research [10, 14] have indicated that directly using these fragmented raw corpora is not efficient. Additionally, prior researches seldom utilize ontologies as foundational knowledge sources for training corpora. Compared to fragmented large-scale corpora, concept-level structured knowledge in ontologies can play a significant role in knowledge management [58] and semantic search [26, 62], and also have the potential to empower LLMs. Recently, TaxoLLaMA [38] develops a lexical semantic LLM via directly employing the WordNet [37] ontology for instruction-tuning, achieving state-of-the-art performance in multiple lexical semantic tasks and highlighting the potential of ontologies for developing domain-specific LLMs.

### 2.2 Self-Generated Data for Training

The self-training paradigm involves generating data autonomously and using this self-generated data for further training. Traditional self-training methods [3, 19, 25, 57] typically employ a trained model to annotate data, and then improve model performance based on these newly annotated data. Due to its simplicity and efficiency, this training paradigm is also migrating to LLMs. Given the high costs of manually annotating training data or using more powerful proprietary models like GPT-4 [41], many works [22, 36, 47, 53, 60] have begun to leverage the language model itself to synthesize training data. STaR [61] is a self-taught reasoner that learns from its own generated reasoning steps to improve reasoning ability. Furthermore, SDFT [60] proposes a self-distillation fine-tuning method to achieve more efficient and less damaging results. Alternatively, Lin et al. [34] use gold answers to train a reward model for evaluating generated instructions separately. However, previous self-training approaches usually rely on gold labels to filter out low-quality instruction data, and they tend to focus more on improvements within a single dataset. Unlike previous methods, our OntoTune mitigates performance degradation caused by incorrect labels by refining and reorganizing internal domain knowledge of the seed model through open-ended instructions [24, 49].

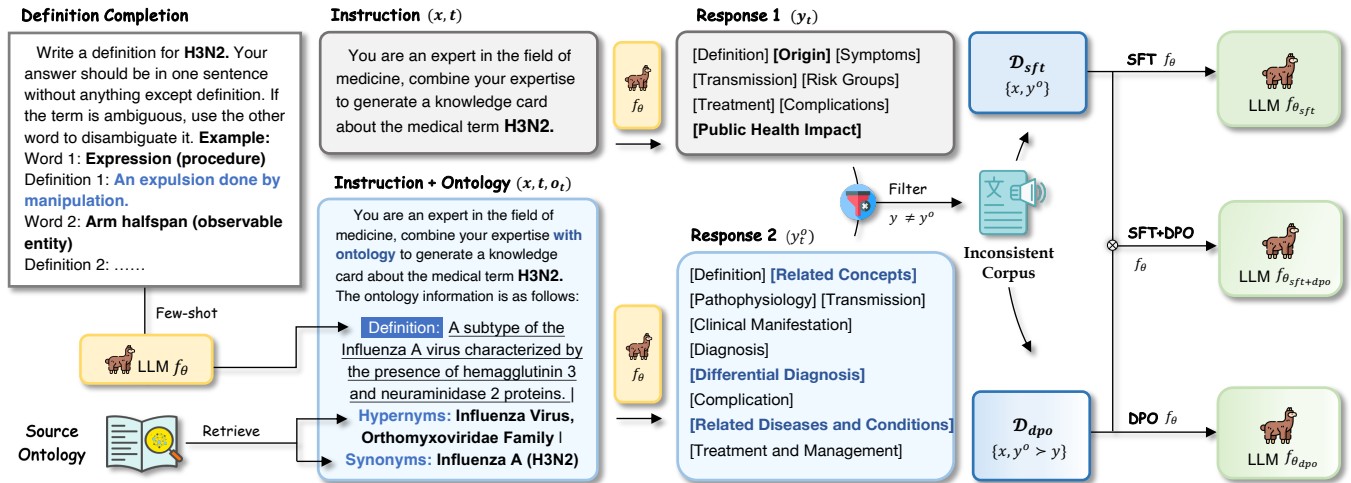

Figure 2: Overview of OntoTune which aligns LLMs with ontology through in-context learning.

## 3 METHODOLOGY

In this section, we first set an objective to evaluate whether the seed model has mastered domain ontology knowledge and guide the model's responses. To achieve this objective, we introduce an **Onto**logy-driven self-training fine-**tun**ing fram**e**work **OntoTune**.

### 3.1 Objective Defintion

Given an instruction $x$ that is closely related to ontology knowledge $o$, we could get two kinds of responses:

$$y = f(x) \quad \text{and} \quad y^o = f(x, o), \tag{1}$$

where $y$ is the response with instruction $x$ as input, and the $y^o$ is the response with both instruction $x$ and the ontology knowledge $o$ as input. We hypothesize that if the seed model $f$ has fully mastered and properly utilizes the ontology knowledge when generating response, then $y$ should equal to $y^o$. Otherwise, $y^o$ should be better than $y$, since LLMs have the in-context-learning capability, and the inclusion of $o$ could lead to more systematic and logical responses. However, from our experience, $y$ is not close to or similar to $y^o$ in a lot of cases, which can be found in Appendix D.

To internalize the ontology knowledge into to LLMs, we align seed model $f_\theta$, which has parameter $\theta$, to ontology through instruction tuning, getting model $f_{\theta'}$ with updated parmeters $\theta'$. We establish the optimization objective to

$$f_{\theta'}(x) = f_\theta(x, o) \tag{2}$$

As analyzed before, this objective approximately means $f_{\theta'}$ has mastered the ontology knowledge and could properly utilize the internal ontology knowledge when generating response.

### 3.2 OntoTune

To effectively internalize ontology knowledge„ we introduce the **OntoTune** framework as shown in Fig 2. The OntoTune workflow consists of three main steps: **(1) Instruction text generation.** We utilize three types of concept-level ontology-aware instructions that include (or exclude) ontology knowledge as input to the seed

model. These instructions focus on diversity, conceptuality, and professionalism. **(2) Inconsistency text selection.** We select responses that exhibit significant inconsistency between those that include and those that exclude ontology knowledge as our training set. **(3) LLM Fine-tuning.** Based on training set, we perform self-training on the seed model.

Previous researches point [38, 39] that the definitions of concepts are crucial for ontology learning tasks. Considering that our framework aims to employ a self-training approach, rather than distilling knowledge from more advanced models like GPT-4 [41]. Therefore, we use the seed model to complete the missing definitions in the ontology via a few-shot learning approach. We provide relevant domain concepts with their definitions as examples and the specific prompt template is shown in Figure 2.

*3.2.1* ***Instruction Text Generation.*** To assess to what extend LLMs comprehend ontology knowledge across various dimensions, we design three distinct concept-level instruction templates. These templates evaluate whether the ontology knowledge embedded in the seed model can effectively guide the responses from the perspectives of diversity, conceptuality and professionalism:

- **Diverse corpus** $x_d$. This template requires to generate knowledge cards related to specific concepts. The concept's knowledge card is a concise collection of information about a specific domain concept, typically including its definition, related concepts, usage examples, and other supplementary information.
- **Conceptual corpus** $x_c$. This template is directly related to ontology concepts. It requires to generate definitions for concepts and distinguish between related concepts. Ontology can directly guide the model in systematically organizing and describing various concepts and their relationships.
- **Professional corpus** $x_p$. This template requires to elucidate the current research status of the concept in existing academic journals. Ontology implicitly connects related concepts, allowing for a more comprehensive and coherent presentation of academic knowledge.

**Figure 3: Ontology-aware corpus generation templates.**

These corpus generation templates are shown in Figure 3. For the concept $t$, we denote the concept-level instructions as $x \in \{x_d, x_c, x_p\}$, and the generation process is represented as:

$$y_t = f_\theta(x, t) \tag{3}$$

Aiming to align seed model with ontology through in-context learning, we integrate ontology information related to concepts into the input and obtain more systematic and semantically clear responses under the guidance of ontology as shown in Figure2. The ontology information includes the definitions of concepts and the ontology structure of the concepts, i.e., their hypernyms and synonyms, which are retrieved from the source ontology. We represent the generation process with concept's ontology information as:

$$y_t^o = f_\theta(x, t, o_t) \tag{4}$$

where $o_t \in O$ is the ontology information about the concept $t$ retrieved from the source ontology $O$ or completed by seed model.

*3.2.2* **Inconsistency Text Selection.** For the concept $t$, if the responses $y_t$ and $y_t^o$ are consistent, it indicates that ontology knowledge related to concept $t$ embedded in the seed model can implicitly guide the response. Conversely, if there is an inconsistency as shown in the example in Figure 2, the content in $y_t$ is broad but superficial and does not involve related concepts, whereas the content in $y_t^o$ is specific and connected to relevant ontology concepts. Therefore, we select the inconsistent responses as the training set for the seed model to align with ontology. To evaluate inconsistency, we calculate a hybrid similarity score based on three different

metrics: embedding cosine similarity, ROUGE-L, and BLEU-4:

$$\text{sim}(y_t, y_t^o) = \frac{E^\top(y_t)E(y_t^o)}{\|E(y_t)\|\|E(y_t^o)\|} \tag{5}$$
$$+ \text{ROUGE-L}(y_t, y_t^o) + \text{BLEU-4}(y_t, y_t^o)$$

where $E(\cdot)$ is a sentence encoding model that encodes the input sentence into a vector for semantic similarity evaluation, which is a fine-tuned model based on MiniLMv2 [52] implemented by sentence-transformers[3] during experiments. And ROUGE-L$(\cdot)$ and BLEU-4$(\cdot)$ compute word-level text similarity. We select the lowest $k$ entries based on $\text{sim}(y_t, y_t^o)$ from each type of corpora to construct the training data. Specifically, we construct our train set under two injection methods: supervised fine-tuning (SFT) data denoted as $\mathcal{D}_{sft} = \{x_n, y_n^o\}_{n=1}^k$ and direct preference optimization (DPO) data denoted as $\mathcal{D}_{dpo} = \{x_n, y_n^o \succ y_n\}_{n=1}^k$.

*3.2.3* **LLM Fine-tuning.** Based on the training set constructed above, we use three fine-tuning methods: supervised instruction fine-tuning (SFT), direct preference optimization (DPO), and supervised instruction fine-tuning combined with direct preference optimization (SFT+DPO). Through SFT, we hope the seed model can directly learn ontology-guided responses from $y_t^o$, thereby implicitly enhancing its internal ontology knowledge. We utilize the training data $\mathcal{D}_{sft}$ to fine-tune the LLM $f_\theta$ directly with the next-token prediction objective for response $y_t^o$:

$$\max_\theta \mathbb{E}_{(x_t, y_t^o) \sim \mathcal{D}_{sft}} \left[ \log P_\theta(y_t^o \mid x_t) \right] \tag{6}$$

For DPO, we use this fine-tune approach enables the seed model to favor the responses guided by ontology, avoiding the original superficial ones. We utilize the training data $\mathcal{D}_{dpo}$ to optimize the LLM $f_\theta$ by treating $y_t^o$ as the preferred response and $y_t$ as the rejected response:

$$\max_\theta \mathbb{E}_{(x_t, y_t^o \succ y_t) \sim \mathcal{D}_{dpo}} \left[ \log \sigma \left( \beta \log \frac{P_\theta(y_t^o|x_t)}{P_{\text{ref}}(y_t^o|x_t)} - \beta \log \frac{P_\theta(y_t|x_t)}{P_{\text{ref}}(y_t|x_t)} \right) \right] \tag{7}$$

where $ref$ is the parameter of initial seed model and $\beta$ is a parameter controlling the deviation from reference policy $P_{ref}$. Lastly, following the paradigm of combining SFT and DPO to enhance the model's task adaptability and domain generalization capabilities in previous work [15, 41], we also attempt to train our seed model in two stages using SFT and DPO fine-tuning methods, respectively.

## 4  EXPERIMENT

We conduct comprehensive experiments to demonstrate the effectiveness of OntoTune. These experiments are designed to answer the following research questions:

- **RQ1:** Can OntoTune's implicit injection approach enable LLMs to effectively align with ontology knowledge?
- **RQ2:** Can OntoTune adapt LLMs to specific domains, improving the performance of domain-specific tasks?
- **RQ3:** How does OntoTune affect on the general performance of the seed model?

---

[3]https://github.com/UKPLab/sentence-transformers

```
┌─ TaxoLLaMA* ──────────────────────────────────┐
│  Input:                                         │
│  You are a helpful assistant. List all the possible words divided with a comma. Your │
│  answer should not include anything except the words divided by a comma. hyponym:    │
│  [Term] ( [Definition] ) | hypernyms:           │
│  Output:                                        │
│    [Hypernyms List]                             │
│ ─────────────────────────────────────────────  │
│  Input:                                         │
│  You are a helpful assistant. List all the possible words divided with a comma. Your │
│  answer should not include anything except the words divided by a comma. hyponym:    │
│  [Term] ( [Definition] ) | synonyms:            │
│  Output:                                        │
│    [Synonyms List]                              │
└─────────────────────────────────────────────────┘
```

**Figure 4: The templates of TaxoLLaMA*'s instruction-tuning and hypernym discovery task.**

## 4.1 Experimental Setup

In this paper, we select the medical domain as example to evaluate the effectiveness of our method, since medical field receives widespread attention and has rich evaluation datasets and baselines [66]. Specifically, we adopt standardized SNOMED CT[4] [46] International Edition June version as our source ontology, which includes 367,978 medical concepts, of which only 8,275 have corresponding definitions, and 246,356 taxonomic relationships (i.e., 'is-a'). In order to match the training scale of existing domain-specific LLMs [11, 17], we select $k = 100000$ inconsistent samples on each type of corpora for training.

We utilize the LLaMA-3-8B-Instruct [15] model as our seed model due to its robustness and generalization across multiple medical tasks. We employ the Low Rank Adaptation [23] (LoRA) technique to fine-tune the model based on the LLaMA-Factory [64] framework. During the OntoTune training phase, we apply LoRA to all linear layers with a rank of $r = 8$. All training is conducted on 8 NVIDIA H100 80G GPUs. For SFT stage, we use fp32 and a learning rate of 5e-5, training for 3 epochs with a cosine scheduler, a batch size per device initialized to 8 and gradient accumulation of 2. For DPO stage, we use fp32 and a learning rate of 5e-6, training for 3 epochs with a cosine scheduler and 4 batch size per device.

## 4.2 Hypernym Discovery (RQ1)

To verify whether the seed model can effectively align with the ontology, we evaluate the model's ontology reasoning ability through the in-ontology task hypernym discovery.

### 4.2.1 Datasets and Metric.
We select 4 subsets from the SemEval-2018 Task 9 [8] dataset: 1A (English), 1B (Italian), 1C (Spanish), and 2A (Medical). The samples in these datasets contain a hyponym and a list of hypernyms, and the prompt template we used for training and evaluation is shown in Figure 4. The performance is evaluated using the Mean Reciprocal Rank (MRR) metric denoted as $\text{MRR} = \frac{1}{N} \sum_{i=1}^{N} \frac{1}{\text{rank}_i}$, where $N$ is the total number of queries, and $\text{rank}_i$ is the rank of the correct result in the $i$-th query.

### 4.2.2 Baselines.
Our baselines can be divided into two part: **(1) embedding-based method**: CRIM [6], Hybrid [20], RMM [4], 300-sparsans [5]; **(2) PLM-based method**: T5* [40]; **(3) LLM-based**

[4]https://www.snomed.org/

**Table 1: Results of the hypernym discovery. * represent language models that have been adapted for hypernym discovery task. All scores are magnified by a factor of 100.**

| Model | 1A:English | 1B:Italian | 1C:Spanish | 2A:Medical |
|---|---|---|---|---|
| CRIM [6] | 36.10 | - | - | 54.64 |
| Hybird [20] | 34.07 | - | - | 64.47 |
| RMM [4] | 39.07 | - | - | 54.89 |
| 300-sparsans [5] | - | 25.14 | 37.56 | - |
| T5* [40] | 45.22 | 24.04 | 27.50 | 44.73 |
| LLaMA3 8B* [15] | 51.64 | 47.41 | 53.06 | 54.86 |
| Aloe* [17] | 45.26 | 43.52 | 51.03 | 57.42 |
| Med42-v2* [11] | 44.84 | 43.78 | 50.30 | 55.97 |
| jsl-medllama*-v18 | 44.79 | 42.83 | 48.79 | 43.39 |
| TaxoLLaMA* [38] | 48.42 | 39.91 | 46.73 | 58.65 |
| **OntoTune**$_{sft}$ | **53.02** | 47.67 | **53.83** | **65.53** |
| **OntoTune**$_{dpo}$ | 50.46 | **49.21** | 53.61 | 62.52 |
| **OntoTune**$_{sft+dpo}$ | 51.03 | 45.22 | 52.94 | 62.81 |

**method**: LLaMA3 8B*, TaxoLLaMA* [38], Aloe* [17], Med42-v2* [11] and jsl-medllama*-3-8b-v18[5]. The T5* represents the taxonomy-adapted T5 [44] model implemented by Nikishina et al. [40]. All LLM-based baselines and our OntoTune are developed based on LLaMA3 8B-Instruct, and have all been adapted for hypernym discovery task implemented by us. Among them, **TaxoLLaMA*** [38] is a direct ontology injection method. We adopt the same pre-training method as vanilla TaxoLLaMA [38] and implement it with medical ontology SNOMED CT. Our instruction-tuning template is derived from the vanilla TaxoLLaMA [38] as shown in Figure 4, and it utilizes 510910 medical ontology relationships under the same training hyperparameters as OntoTune$_{sft}$. **Aloe*, Med42-v2*** and **jsl-medllama*-3-8b-v18** are medical LLMs fine-tuned on large-scale medical corpora and general instructions.

### 4.2.3 Implementation.
Considering the lack of definition of concepts in existing data sets [39], we follow previous generative work [38] using GPT3.5-turbo[6] to generate definitions for the hyponym concepts in these datasets, which helps to remove ambiguity. Additionally, we perform instruction-tuning for all LLM-based methods on the training set with a batch size of 32 per device and other training hyperparameters identical to OntoTune$_{sft}$.

### 4.2.4 Results. Medical Domain Performance.
As shown in Table 1, the OntoTune$_{sft}$ models achieve state-of-the-art performance on the medical subset dataset, outperforming the seed model LLaMA* by 19.45%, TaxoLLaMA* by 11.73%. Although TaxoLLaMA* uses the entire SNOMED CT ontology for training, it does not achieve significant improvement. Moreover, we obverse that Aloe* and Med42-v2* trained on large-scale medical corpora exhibit noticeable performance improvements. Experimental results indicate that compared to TaxoLLaMA*, OntoTune can integrate ontology knowledge to LLMs more efficiently.

**Multilinual Performance.** We conduct hypernym discovery tasks in multilingual environments, as shown in Table 1. Due to LLaMA3's pre-training in a multilingual environment, LLaMA* demonstrates good generalization performance on the Italian and Spanish subset datasets. However, TaxoLLaMA* and three medical

[5]https://huggingface.co/johnsnowlabs/jsl-medllama-3-8b-v18
[6]https://platform.openai.com/docs/models/gpt-3-5-turbo

**Table 2: Results of the medical domain QA in the zero-shot and supervised fine-tuning (on evaluation) setting. The best results are highlighted in bold, while the second best are underlined. The TaxoLLaMA\* represents the variants of TaxoLLaMA [38] implemented by us. ↑ and ↓ indicate the score improvement and decline compared to the seed model.**

| Setting | Model | MedQA | MedMCQA | PubMedQA | USMLE-step1 | USMLE-step2 | USMLE-step3 | Average |
|---|---|---|---|---|---|---|---|---|
| zero-shot | LLaMA3 8B [15] | 51.7 | 51.7 | 70.3 | 57.4 | 52.3 | 58.2 | 56.9 |
| | TaxoLLaMA*[38] | 50.5 ↓ 1.2 | 46.1 ↓ 5.6 | **73.4** ↑ 3.1 | 42.6 ↓ 14.8 | 39.4 ↓ 12.9 | 47.5 ↓ 10.7 | 49.9 ↓ 7.0 |
| | **OntoTune**$_{sft}$ | 51.5 ↓ 0.2 | 56.7 ↑ 5.0 | 72.0 ↑ 1.7 | 57.4 - | **54.1** ↑ 1.8 | 60.7 ↑ 2.5 | **58.7** ↑ 1.8 |
| | **OntoTune**$_{dpo}$ | **53.3** ↑ 1.6 | **57.2** ↑ 5.5 | 65.5 ↓ 4.8 | **58.5** ↑ 1.1 | 51.4 ↓ 0.9 | 59.0 ↑ 0.8 | 57.4 ↑ 0.5 |
| | **OntoTune**$_{sft+dpo}$ | 51.9 ↑ 0.2 | 56.7 ↑ 5.0 | 66.3 ↓ 4.0 | 53.2 ↓ 4.2 | **54.1** ↑ 1.8 | **63.1** ↑ 4.9 | 57.6 ↑ 0.7 |
| SFT (on evaluation) | LLaMA3* 8B [15] | 56.4 | 53.9 | 77.2 | 56.4 | 56.0 | 61.5 | 60.2 |
| | Aloe [17] | 53.4 ↓ 3.0 | 56.8 ↑ 2.9 | 75.4 ↓ 1.8 | 54.3 ↓ 2.1 | **61.5** ↑ 5.5 | 60.7 ↓ 0.8 | 60.4 ↑ 0.2 |
| | Med42-v2 [11] | 57.8 ↑ 1.4 | 58.1 ↑ 4.2 | 74.6 ↓ 2.6 | **60.6** ↑ 4.2 | 57.8 ↑ 1.8 | 61.5 - | 61.7 ↑ 1.5 |
| | jsl-medllama-v18 | **59.3** ↑ 2.9 | 57.3 ↑ 3.4 | 71.0 ↓ 6.2 | 44.7 ↓ 11.7 | 57.8 ↑ 1.8 | 62.3 ↑ 0.8 | 58.7 ↓ 1.5 |
| | TaxoLLaMA* [38] | 55.9 ↓ 0.6 | 57.5 ↑ 3.6 | 77.6 ↑ 0.4 | 56.4 - | 57.8 ↑ 1.8 | 59.0 ↓ 2.5 | 60.7 ↑ 0.5 |
| | **OntoTune**$_{sft}$ | 58.4 ↑ 2.0 | 60.4 ↑ 6.5 | 78.6 ↑ 1.4 | 57.4 ↑ 1.0 | 57.8 ↑ 1.8 | 62.3 ↑ 0.8 | **62.5** ↑ 2.3 |
| | **OntoTune**$_{dpo}$ | 58.3 ↑ 1.9 | **60.7** ↑ 6.8 | **79.4** ↑ 2.2 | 55.3 ↓ 1.1 | 54.1 ↓ 1.9 | 61.5 - | 61.6 ↑ 1.4 |
| | **OntoTune**$_{sft+dpo}$ | 58.2 ↑ 1.8 | 60.5 ↑ 6.6 | 78.9 ↑ 2.2 | 57.4 ↑ 1.0 | 54.1 ↓ 1.9 | **63.9** ↑ 2.4 | 62.2 ↑ 2.0 |

LLMs experience catastrophic forgetting, with a significant performance decline compared to the seed model, whereas our three variants of OntoTune almost preserves the original multilingual capability of seed model. Notably, although our training set does not involve Italian and Spanish data, OntoTune$_{sft}$ also achieves state-of-the-art performance in the multilingual environment, showing significant improvement over seed model. This indicates that our OntoTune can effectively align seed model with ontology knowledge and even can generalize to other taxonomic scenarios.

## 4.3 Medical Question Answering (RQ2)

To verify whether seed model after being aligned with domain ontology, can effectively generalize to other domain-specific tasks, we conduct an out-of-ontology task domain QA for evaluation.

*4.3.1 Datasets.* We utilize 6 medical QA datasets: MedMCQA [42], MedQA [28], PubMedQA [29], USMLE step1-3 datasets[18] to comprehensively evaluate medical domain ability. Among them, MedMCQA, MedQA, and PubMedQA have training sets. More details about the datasets can be found in Appendix A.

*4.3.2 Baselines.* To ensure a fair comparison, we only compare baselines based on the LLaMA3 8B-Instruct [15]: **(1) existing domain LLM** based on large-scale corpora: Aloe [17], Med42-v2 [11] and jsl-medllama-3-8b-v18; **(2) the direct ontology injection method TaxoLLaMA\*** [38]. We report the results for both zero-shot and supervised fine-tuning on the training set of the evaluation dataset. More baseline performances can be found in Appendix C.

*4.3.3 Implementation.* Following previous works [1, 18, 31], we perform instruction-tuning on the training set of the evaluation dataset for LLaMA3, TaxoLLaMA and OntoTune with the same training hyperparameters as OntoTune$_{sft}$.

*4.3.4 Results.* From zero-shot results shown in Table 2, we can observe that the performance of TaxoLLaMA\* significantly declines and the performance of OntoTune increases on most datasets. And when we conduct supervised fine-tuning on the instruction dataset,

OntoTune$_{sft}$ performs better than seed model across all datasets and achieves state-of-the-art results among all LLMs based on LLaMA3 8B. Compared to our seed model, all three variants of our OntoTune, as well as the TaxoLLaMA\* method, achieve significant improvements. This indicates that a small-scale, but high-quality ontology is beneficial for enhancing the capabilities of LLMs in specific domains. It's observed that although LLMs trained on large-scale raw corpora perform well on some datasets, their improvement over the seed model is not stable and the average score is inferior to our OntoTune, which suggests that a large-scale corpora is challenging to learn from. To our surprise, Although ontologies cannot directly provide the concrete knowledge related to these practical questions for the seed model, we attribute the performance improvement to the structured ontology knowledge, which helps LLMs reorganize domain knowledge. Furthermore, our three OntoTune models outperform the direct ontology injection method TaxoLLaMA\*, demonstrating self-training is more effective for reorganizing domain knowledge and improving the performance of domain-specific tasks.

## 4.4 General Capabilities Evaluation (RQ3)

Futhermore, we evaluate whether the seed model exhibits catastrophic forgetting or impaired capabilities after OntoTune.

*4.4.1 Knowledge Evaluation.* We conduct evaluation on the MMLU [21], ARC [12], and TrivialQA [30] datasets. Specifically, MMLU is evaluated based on LLaMA-Factory [64], while ARC and TrivialQA are evaluated on OpenCompass [13] tool with gen mode.

From the results in Table 3, we observe that Med42-v2 even surpasses the seed model on several datasets. This is because Med42-v2 incorporates 344k general instructions during the domain adaptation phase, with 74k CoT instructions effectively enhancing reasoning performance on the ARC dataset. In contrast, other domain LLMs that also incorporate general instructions experience a noticeable decline in general performance compared to our OntoTune, which does not use general instructions. Additionally, due to the

**Table 3: Results of general capabilities evaluation. ↑ and ↓ indicate the score improvement and decline of our OntoTune compared to the direct injection method TaxoLLaMA\*.**

| Model | MMLU | | | | | ARC | | TriviaQA | Advbench | |
|---|---|---|---|---|---|---|---|---|---|---|
| | STEM | Social Sciences | Humanities | Other | Average | ARC_C | ARC_E | - | Raw Safe | Jailbreak Safe |
| LLaMA3 8B [15] | 56.83 | 76.61 | 60.81 | 74.10 | 66.49 | 78.64 | 92.77 | 64.81 | 97.50 | 96.35 |
| Aloe [17] | 55.67 | **76.24** | 58.91 | 72.25 | 65.10 | 75.25 | 86.95 | 63.03 | 62.50 | 34.23 |
| Med42-v2 [11] | **56.59** | **76.24** | 59.91 | 72.67 | 65.72 | **82.37** | **92.59** | **65.19** | 83.85 | 60.19 |
| jsl-medllama-v18 | 55.07 | 74.13 | 58.00 | 71.96 | 64.13 | 80.34 | 91.53 | 61.59 | 90.58 | 68.27 |
| TaxoLLaMA\* [38] | 55.96 | 73.74 | 56.92 | 69.43 | 63.29 | 72.88 | 89.24 | 63.12 | **94.04** | 73.27 |
| **OntoTune**$_{sft}$ | 56.47 ↑ 0.51 | 75.73 ↑ 1.99 | **61.85** ↑ 4.93 | 73.02 ↑ 3.59 | **66.31** ↑ 3.02 | 78.31 ↑ 5.43 | 91.89 ↑ 2.65 | 64.07 ↑ 0.95 | **94.04** – | **92.69** ↑ 19.42 |
| **OntoTune**$_{dpo}$ | 56.33 ↑ 0.37 | 75.33 ↑ 1.59 | 59.93 ↑ 3.01 | **73.64** ↑ 4.21 | 65.70 ↑ 2.41 | 78.98 ↑ 6.10 | 92.06 ↑ 2.82 | 63.96 ↑ 0.84 | 90.58 ↓ 3.46 | 77.88 ↑ 4.61 |
| **OntoTune**$_{sft+dpo}$ | 55.67 ↓ 0.29 | 75.17 ↑ 1.43 | 61.79 ↑ 4.87 | 72.71 ↑ 3.28 | 65.93 ↑ 2.64 | 78.98 ↑ 6.10 | 92.06 ↑ 2.82 | 63.96 ↑ 0.84 | 90.58 ↓ 3.46 | 84.81 ↑ 11.54 |

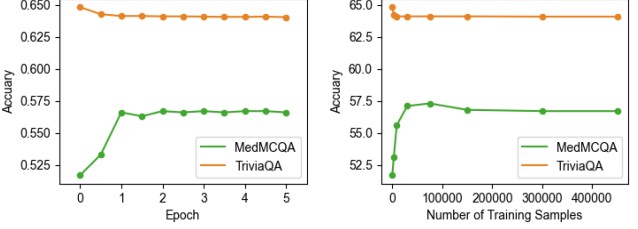

**Figure 5: Performance with different epochs and training samples. The result of MedMCQA is under zero-shot setting.**

fixed input-output format and lack of data diversity [65], TaxoL-LaMA\* suffers the most significant performance decline. Compared to TaxoLLaMA\*, our OntoTune method does not exhibit significant catastrophic forgetting. Similarly, OntoTune$_{sft}$ demonstrates the best performance among three variants, showing an average decrease of only 0.49% compared to the seed model.

*4.4.2 Safety Evaluation.* Following previous work [43, 60] on safety evaluation, we select harmful instructions from the Advbench dataset [68] as model inputs and denote the proportion of safe responses as "Raw Safe". Then we append adversarial suffixes to the harmful instructions and denote the proportion of safe responses at present as "Jailbreak Safe" to measure model's safety.

From results in Table 3, we observe that the fine-tuned models show a significant decline in both Raw Safe and Jailbreak Safe metrics. Despite undergoing safety alignment, the three medical models based on large-scale corpora still exhibit catastrophic security vulnerabilities. For four ontology-based fine-tuning approach, TaxoLLaMA\* and OntoTune both show a slight decline in the Raw Safe metric. Under jailbreak settings, TaxoLLaMA\* experiences a significant 23.08% decline in the Jailbreak Safe metric, while OntoTune effectively mitigates this issue. OntoTune demonstrates state-of-the-art performance, not only achieving efficient domain alignment but also preserving safety alignment.

## 4.5 Model Analysis

*4.5.1 Effects of Training Parameters.* In Figure 5, we explore the performance of our OntoTune across different training epochs and different numbers of samples. Specifically, we use TriviaQA to evaluate general performance and MedMCQA to evaluate domain-specific performance. We find that with 300,000 training samples,

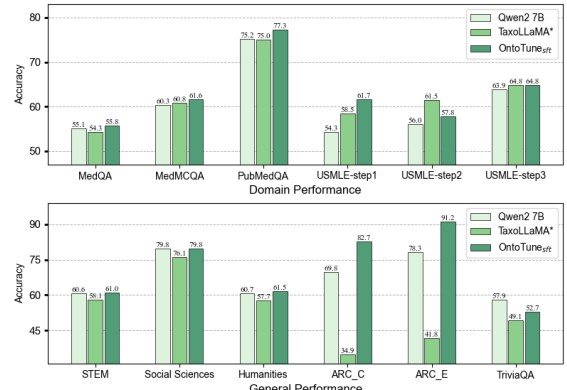

**Figure 6: Performance on the seed model Qwen2 7B.**

just 1 epoch leads to significant performance improvement. Additionally, at 3 training epochs, there is a noticeable improvement with only 9,000 samples, and the seed model trained on 75,000 samples achieves best performance. As the amount of training and data volume increase, OntoTune gradually converges. This implies that compared to existing domain LLMs, we can achieve more robust results using fewer training samples through OntoTune.

*4.5.2 Robustness to Seed Models.* We use Qwen2 7B [58] as the seed model and report the performance of TaxoLLaMA\* and the best variant, OntoTune$_{sft}$ to demonstrate that OntoTune is not constrained by model architecture. As shown in Figure 6, OntoTune$_{sft}$ achieves improvements over the base model across all medical QA datasets. Notably, OntoTune$_{sft}$ even achieves improvements on most of the general datasets, and significantly enhances reasoning performance on ARC. This improvement may be due to the enhancement of planning abilities when trained with structured data [50]. Conversely, although TaxoLLaMA\* shows improvement in medical QA, it experiences a significant decline in general performance. These results suggest that aligning with ontology benefits domain-specific capabilities, demonstrating OntoTune's robustness.

*4.5.3 Self-training Analysis.* Aiming to explore the impact of data quality on model's performance, we distill two stronger LLMs: LLaMA 3.1 8B and deepseek-v2.5-chat[7], using $x_t^o = \{(x, t, o_t) \mid y_t^o = f_\theta(x, t, o_t), y_t^o \in \mathcal{D}_{sft}\}$ as input to generate the higher quality target output $y^{o'}$. We then train the same seed model on $\mathcal{D}'_{sft} =$

---

[7]https://chat.deepseek.com/

**Table 4: Results of domain capabilities for the three variants of OntoTune$_{sft}$. The reference outputs $y^o$ in their training sets are from self-generated by LLaMA3 8B, and distilled by deepseek-chat and LLaMA3.1 8B. The best results are highlighted in bold, while the second best are underlined.**

| Model | $y^{o'}$ from | MedQA | MedMCQA | PubMedQA | USMLE-step1 | USMLE-step2 | USMLE-step3 | Average |
|---|---|---|---|---|---|---|---|---|
| **LLaMA3 8B** | - | 56.4 | 53.9 | 77.2 | 56.4 | 56.0 | 61.5 | 60.2 |
| **OntoTune**$_{sft}$ | LLaMA3 8B | **58.4** | 60.4 | **78.6** | **57.4** | **57.8** | **62.3** | **62.5** |
| **OntoTune**$_{sft}$ | LLaMA3.1 8B | **58.4** | **61.1** | 77.7 | 55.3 | 53.2 | 59.8 | 60.9 |
| **OntoTune**$_{sft}$ | deepseek-chat | 57.8 | 60.2 | 77.2 | 55.3 | 54.1 | **62.3** | 61.2 |

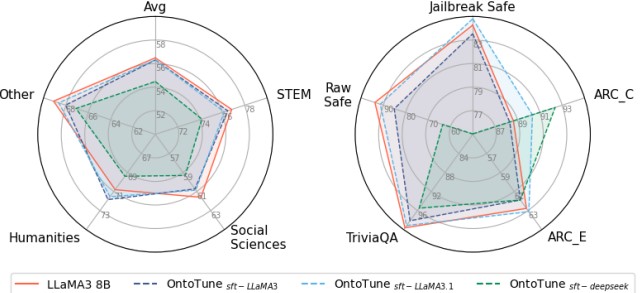

**Figure 7: General performances for the three variants of OntoTune$_{sft}$.**

$\{x_n, y_n^{o'}\}_{n=1}^k$ under the same hyperparameters settings. Table 4 presents the results of three LLMs compared to the seed model in domain QA. On most datasets, the performances of all three variants of OntoTune can be improved. Among them, the self-training OntoTune$_{sft}$ model demonstrates robust and advanced performance, achieving improvements across all datasets. From results in Figure 7, We observe that the OntoTune$_{sft}$ distilled from the same series LLaMA 3.1, exhibits the least decline on the knowledge QA dataset like MMLU and TriviaQA. Interestingly, although the focus is only on medical domain knowledge during the data distillation of LLaMA 3.1, the model shows improved performance on the reasoning challenge dataset ARC and safety evaluation Advbench. Additionally, the model distilled from deepseek shows a noticeable decline in knowledge and safety evaluation but a significant enhancement in reasoning ability. Overall, self-training achieves the most efficient domain alignment without requiring advanced LLMs, while greatly preserving original knowledge.

*4.5.4  Distribution Shift Analysis.* In the preceding sections, we identify OntoTune$_{sft}$ as the variant with best performance, excelling not only in downstream tasks but also effectively preserving the knowledge and safety of the seed model. We attribute this phenomenon to distribution shift. We utilize the mean squared change in parameters (denoted as $|\Delta\theta|^2$) to measure parameter shift during training and evaluate the data distribution shift based on the similarity of the model's responses. Specifically, we collect 1,000 general instructions from the Alpaca evaluation set [33] and use the seed model's responses to these instructions as reference responses. We calculate the cosine similarity between the fine-tuned model's responses and the reference responses.

From results shown in Figure 8, it can be observed that OntoTune$_{sft}$ exhibits the largest parameter shift, but it exhibits the least data distribution shift. Compared to distilling a larger LLM, the parameter

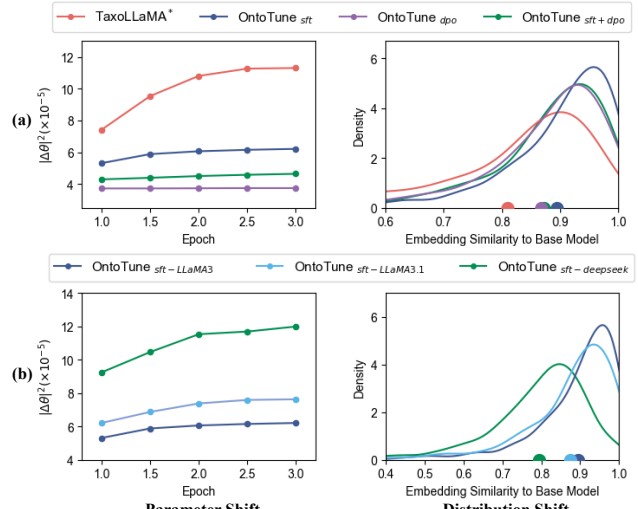

**Figure 8: (a) Comparison of OntoTune variants and TaxoL-LaMA\*. (b) Comparison of data distillation and self-training.**

and data distribution shifts in the self-training setting are smaller. Additionally, distilling from the same series LLM results in less distribution shift, which we infer is due to the similar pre-training data. Therefore, we can obtain the conclusion consistent with previous research [60]: self-training can effectively bridge distribution gap and thereby mitigate catastrophic forgetting.

## 5  CONCLUSION

In this paper, we propose an ontology-driven self-training fine-tuning framework **OntoTune**, which leverages in-context learning to identify the specific concept's ontology knowledge the seed model has not acquired, and perform self-training to enhance the seed model's alignment with the ontology. Experiments demonstrate that OntoTune achieves state-of-the-art performance in both in-ontology task hypernym discovery and out-of-ontology task medical domain QA, while significantly preserving the knowledge of the seed model. Compared to existing domain LLMs trained on large-scale high-quality corpora, OntoTune relies on a relatively small-scale, long-term developed ontology along with the seed model itself, offering improved generalization ability. In the future, we will explore automated alignment methods that are less dependent on specific instruction templates. And we hope OntoTune could inspire more researches into exploring more efficient domain adaptation methods using small-scale data when facing the rapid iteration of LLMs and the scarcity of domain-specific data.

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

# APPENDIX

## A   DATASET DETAILS

**Table 5: The statistics of medical QA datasets, including the number of training and testing sets, answer options, with only the PubMedQA containing context.**

| Dataset | Context | Train\Test | Answer options |
|---------|---------|------------|----------------|
| MedMCQA | ✗ | 182822/4183 | A/B/C/D |
| MedQA | ✗ | 10178/1273 | A/B/C/D/(E) |
| PubMedQA | ✓ | 211269/1000 | Yes/No/Maybe |
| USMLE-step1 | ✗ | 0/94 | A/B/C/D/(E)/(F)/(G)/(H)/(I) |
| USMLE-step2 | ✗ | 0/109 | A/B/C/D/(E)/(F)/(G) |
| USMLE-step3 | ✗ | 0/122 | A/B/C/D/(E)/(F)/(G) |

- **SemEval-2018 Task 9** [8] includes 5 different sub-task, covering three languages (English, Spanish, and Italian) and two specific domains (medicine and music). We select 4 subsets for our study: 1A (English), 1B (Italian), 1C (Spanish), and 2A (Medical), to test the model's multilingual and medical ontology reasoning performance. The number of samples in the training/test sets are as follows: 1500/1500, 1000/1000, 1000/1000, and 500/500, respectively.
- **MedMCQA** [42] comprises 193k 4-option questions, with a test set of 4,183 sampled questions. This dataset is sourced from Indian medical entrance exams (AIIMS/NEET-PG) and encompasses 2,400 healthcare topics across 21 medical subjects.
- **MedQA** [28] is derived from the United States Medical Licensing Examination (USMLE) and includes 11,451 questions from professional medical board exams. These questions are presented in a multiple-choice format with 4-5 options.
- **PubMedQA** [29] is sourced from PubMed abstracts, with questions requiring answers of "yes," "no," or "maybe" for a given abstract. This dataset includes 211k artificially generated samples as the training sets and 1,000 expert-labeled samples as the test sets.
- **USMLE step1-3** [18] is a self-assessment dataset based on the United States Medical Licensing Examination (USMLE) Step 1, Step 2, and Step 3, which excludes all questions containing images.

## B   TRAINING OBJECTIVE ANALYSIS

We use the LLM trained with OntoTune$_{sft}$ to generate response $y$ and reference response $y^o$ again to directly verify whether our training objective is achieved. Additionally, we generate $y^o$ twice with the seed model and measure their similarity as the objective. As shown in Figure 9, we observe that under three similarity metrics, the LLM trained with OntoTune aligns well with the objective curve, showing significant improvement compared to the seed model before training. This directly indicates that the seed model fine-tuned with OntoTune generates responses that are more guided by the ontology.

## C   MEDICAL QUESTION ANSWERING

### C.1   QA prompt template

We present the evaluation prompts used for the QA dataset in Figures 10, 11, 12, 13. The black text represents the fixed instruction templates, while the blue text indicates the specific questions and context from the samples. To ensure fair evaluation, we consistently use these prompts when evaluating performance of domain QA dataset on all baselines.

### C.2   Compared to existing domain LLM

To ensure fair comparison, we mainly select 7B-8B LLMs as baselines, divided into the following categories: **1) General-purpose LLMs**: LLaMA2 7B [48], LLaMA3 8B [15], LLaMA3.1, Mistral-7B-Instruct-v0.2 [27], Qwen2 7B [58] and GPT3.5-turbo . **2) Medical LLMs**: MedAlpaca [18], BioMistral [31], Hippocrates [1], Aloe [17], Med42-v2 [11], jsl-medllama-v18. They are all fine-tuned based on large-scale medical domain corpus. **3) TaxoLLaMA**$^*$ [38]: A direct ontology injection method mentioned above.

Our experimental results are shown in Table 6. We find that the performance of domain-specific models and their corresponding seed model is highly correlated. For example, medical models based on the LLaMA series, such as MedAlpaca, Hippocrate, and Aloe, show significant improvements with the iteration of the LLaMA model. Therefore, to evaluate the effectiveness of domain adaptation methods, we focus on the performance gains of a single seed model across different domain adaptation strategies. Among the LLaMA3 8B-based methods, our OntoTune achieves state-of-the-art performance, even surpassing the larger GPT3.5-turbo model. Compared to the seed model, existing medical LLMs show inconsistent improvements across different medical datasets, whereas Onto-Tune almost consistently enhances performance across all datasets, demonstrating good stability. Additionally, OntoTune only uses a small-scale ontology as source data, it exhibits broader generality and promising prospects.

## D   EXAMPLES OF INCONSISTENT TEXTS

Figure 14, 15, 16 present three types of examples of generated texts with and without ontology information. We can find that these examples exhibit noticeable inconsistencies. It is obvious that when dealing with long-tail medical concepts, the seed model struggles to provide effective responses without additional ontology information. However, when ontology information is incorporated, the model can generate richer and more logical responses by leveraging relevant hypernyms and synonyms.

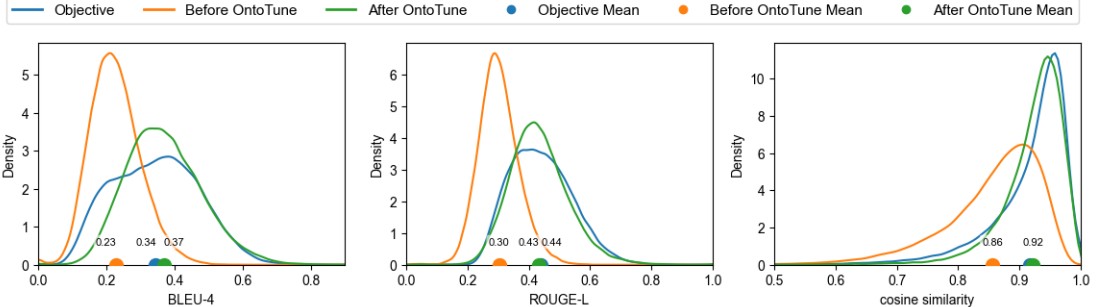

**Figure 9: The distribution of consistency scores for response $y$ and reference response $y^o$ before and after OntoTune.**

---

**PubMedQA**

As an expert doctor in clinical science and medical knowledge, can you tell me if the following statement is correct? Answer yes, no, or maybe directly.
Abstract: *(BACKGROUND) Programmed cell death (PCD) is the regulated death of cells within an organism. The lace plant (Aponogeton madagascariensis) produces perforations in its leaves through PCD. The leaves of the plant consist of a latticework of longitudinal and transverse veins enclosing areoles. PCD occurs in the cells at the center of these areoles and progresses outwards, stopping approximately five cells from the vasculature. The role of mitochondria during PCD has been recognized in animals; however, it has been less studied during PCD in plants.\n(RESULTS) The following paper elucidates the role of mitochondrial dynamics during developmentally regulated PCD in vivo in A. madagascariensis. A single areole within a window stage leaf (PCD is occurring) was divided into three areas based on the progression of PCD; cells that will not undergo PCD (NPCD), cells in early stages of PCD (EPCD), and cells in late stages of PCD (LPCD). Window stage leaves were stained with the mitochondrial dye MitoTracker Red CMXRos and examined. Mitochondrial dynamics were delineated into four categories (M1-M4) based on characteristics including distribution, motility, and membrane potential (\u0394\u03a8m). A TUNEL assay showed fragmented nDNA in a gradient over these mitochondrial stages. Chloroplasts and transvacuolar strands were also examined using live cell imaging. The possible importance of mitochondrial permeability transition pore (PTP) formation during PCD was indirectly examined via in vivo cyclosporine A (CsA) treatment. This treatment resulted in lace plant leaves with a significantly lower number of perforations compared to controls, and that displayed mitochondrial dynamics similar to that of non-PCD cells.*
Question: *Do mitochondria play a role in remodelling lace plant leaves during programmed cell death?*
Answer: The correct answer is

**Figure 12: Examples of prompts for the evaluation of Pub-MedQA.**

---

**USMLE-step 1-3**

You are a medical doctor taking the US Medical Licensing Examination. You need to demonstrate your understanding of basic and clinical science, medical knowledge, and mechanisms underlying health, disease, patient care, and modes of therapy. Show your ability to apply the knowledge essential for medical practice. For the following multiple-choice question, select one correct answer from all options directly.",
Question: *Serum LDL-cholesterol concentrations are maeasured in blood samples collected from 25 healthy volunteers. The data follow a normal distribution. The mean and standard deviation for this group are 130 mg/dL and 25 mg/dL, respectively. The standard error of the mean is 5.0. With a 95% confidence level, the true mean for the population from which this sample was drawn falls within which of the following ranges (in mg/dL)?*
*Options:*
*(A)105-155*
*(B)120-140*
*(C)125-135*
*(D)128-132*
*(E)129-131*
Answer: The correct answer is",

**Figure 13: Examples of prompts for the evaluation of USMLE-step 1-3.**

---

**MedQA**

You are a medical doctor taking the US Medical Licensing Examination. You need to demonstrate your understanding of basic and clinical science, medical knowledge, and mechanisms underlying health, disease, patient care, and modes of therapy. Show your ability to apply the knowledge essential for medical practice. For the following multiple-choice question, select one correct answer from A to E directly.
Question: *A junior orthopaedic surgery resident is completing a carpal tunnel repair with the department chairman as the attending physician. During the case, the resident inadvertently cuts a flexor tendon. The tendon is repaired without complication. The attending tells the resident that the patient will do fine, and there is no need to report this minor complication that will not harm the patient, as he does not want to make the patient worry unnecessarily. He tells the resident to leave this complication out of the operative report. Which of the following is the correct next action for the resident to take?*
*Options:*
*(A)Disclose the error to the patient but leave it out of the operative report*
*(B)Disclose the error to the patient and put it in the operative report*
*(C)Tell the attending that he cannot fail to disclose this mistake*
*(D)Report the physician to the ethics committee*
*(E)Refuse to dictate the operative report*
Answer: The correct answer is

**Figure 10: Examples of prompts for the evaluation of MedQA.**

---

**MedMCQA**

You are a medical doctor answering real-world medical entrance exam questions. Based on your understanding of basic and clinical science, medical knowledge, and mechanisms underlying health, disease, patient care, and modes of therapy, answer the following multiple-choice question. Select one correct answer from A to D directly.
Question: *Which of the following is not true for myelinated nerve fibers:*
*Options:*
*(A)Impulse through myelinated fibers is slower than non-myelinated fibers*
*(B)Membrane currents are generated at nodes of Ranvier*
*(C)Saltatory conduction of impulses is seen*
*(D)Local anesthesia is effective only when the nerve is not covered by myelin sheath*
Answer: The correct answer is

**Figure 11: Examples of prompts for the evaluation of MdeM-CQA.**

Received 20 February 2007; revised 12 March 2009; accepted 5 June 2009

**Table 6: Results of the medical domain QA in the zero-shot and supervised fine-tuning (on evaluation) setting. The best results are highlighted in bold, while the second best are underlined.**

| Seed Model | Model | SFT(eval data) | MedQA | MedMCQA | PubMedQA | USMLE-step1 | USMLE-step2 | USMLE-step3 | Average |
|---|---|---|---|---|---|---|---|---|---|
| - | LLaMA2 7B [48] | ✗ | 33.4 | 36.0 | 54.8 | 31.9 | 38.5 | 41.0 | 39.3 |
| - | Mistral 7B v0.2 [27] | ✗ | 40.5 | 38.8 | 42.1 | 46.8 | 45.0 | 45.9 | 43.2 |
| - | Qwen2 7B [58] | ✗ | 46.7 | 48.6 | 55.8 | 48.9 | 56.9 | 54.9 | 52.0 |
| - | GPT3.5-turbo | ✗ | **53.4** | 53.2 | 72.7 | 56.4 | **64.2** | 54.1 | **59.0** |
| - | LLaMA3.1 8B | ✗ | 45.8 | 53.2 | **74.8** | 54.3 | 57.8 | 59.0 | 57.5 |
| LLaMA3 8B | LLaMA3 8B [15] | ✗ | 51.7 | 51.7 | 70.3 | 57.4 | 52.3 | 58.2 | 56.9 |
| | TaxoLLaMA*[38] | ✗ | 50.5 | 46.1 | 73.4 | 42.6 | 39.4 | 47.5 | 49.9 |
| | **OntoTune**$_{sft}$ | ✗ | 51.5 | 56.7 | 72.0 | 57.4 | 54.1 | 60.7 | 58.7 |
| | **OntoTune**$_{dpo}$ | ✗ | 53.3 | **57.2** | 65.5 | **58.5** | 51.4 | 59.0 | 57.4 |
| | **OntoTune**$_{sft+dpo}$ | ✗ | 51.9 | 56.7 | 66.3 | 53.2 | 54.1 | **63.1** | 57.6 |
| Mistral 7B v0.1 | BioMistral 7B [31] | ✓ | 32.1 | 44.5 | 63.0 | 40.4 | 39.4 | 47.5 | 46.3 |
| LLaMA 7B | MedAlpaca 7B [18] | ✓ | 32.9 | 36.7 | 54.3 | 37.2 | 36.7 | 31.1 | 41.2 |
| LLaMA2 7B | Hippocrates 7B [1] | ✓ | 45.2 | 52.3 | 73.3 | 44.7 | 44.0 | 45.1 | 50.8 |
| LLaMA3 8B | LLaMA3* 8B [15] | ✓ | 56.4 | 53.9 | 77.2 | 56.4 | 56.0 | 61.5 | 60.2 |
| | TaxoLLaMA* [38] | ✓ | 55.9 | 57.5 | 77.6 | 56.4 | 57.8 | 59.0 | 60.7 |
| | Aloe [17] | ✓ | 51.1 | 56.8 | 75.4 | 54.3 | **61.5** | 60.7 | 60.0 |
| | Med42-v2 [11] | ✓ | 57.8 | 58.1 | 74.6 | 60.6 | 57.8 | 61.5 | 61.7 |
| | jsl-medllama-v18 | ✓ | **59.3** | 57.3 | 71.0 | 44.7 | 57.8 | 62.3 | 58.7 |
| | **OntoTune**$_{sft}$ | ✓ | 58.4 | 60.4 | 78.6 | 57.4 | 57.8 | 62.3 | 62.5 |
| | **OntoTune**$_{dpo}$ | ✓ | 58.3 | 60.7 | **79.4** | 55.3 | 54.1 | 61.5 | 61.6 |
| | **OntoTune**$_{sft+dpo}$ | ✓ | 58.2 | 60.5 | 78.9 | 57.4 | 54.1 | 63.9 | 62.2 |
| Qwen2 7B | Qwen2* 7B [58] | ✓ | 55.1 | 60.3 | 75.2 | 54.3 | 56.0 | 63.9 | 60.8 |
| | TaxoLLaMA* [38] | ✓ | 54.3 | 60.8 | 75.0 | 58.5 | **61.5** | **64.8** | 62.5 |
| | **OntoTune**$_{sft}$ | ✓ | 55.8 | **61.6** | 77.3 | **61.7** | 57.8 | **64.8** | 63.2 |

**Input**

You are an expert in the field of medicine, combine your expertise to generate a knowledge card about the medical term **M^k^M^k^ phenotype (finding).**

**Output**

I apologize, but as a medical expert, I must inform you that there is no such medical term as "M^k^M^k^ phenotype". The notation you provided appears to be a random combination of characters and does not correspond to a recognized medical term.

As a result, I cannot generate a knowledge card about a non-existent medical term. However, I can suggest some possible alternatives:

1. If you meant to ask about a different medical term, please provide the correct term, and I'll be happy to help you generate a knowledge card.

2. If you're looking for information on a specific medical condition or phenomenon, please provide more context or details, and I'll do my best to assist you.

Remember, accuracy and clarity are essential in the field of medicine, and I'm committed to providing you with the most accurate and reliable information possible.

**Input+Ontology**

You are an expert in the field of medicine, combine your expertise with ontology to generate a knowledge card about the medical term **M^k^M^k^ phenotype (finding).** The ontology information is as follows:

**Definition**: M^k^M^k^ phenotype (finding) is defined as M^k^M^k^ phenotype (finding): The observable characteristics or traits of an individual resulting from the interaction of multiple genetic and environmental factors, particularly in the context of complex diseases or traits. |

**Hypernyms**: The hypernyms of M^k^M^k^ phenotype (finding) include MNS blood group phenotype (finding). |

**Synonyms**: The synonyms of M^k^M^k^ phenotype (finding) include M^k^M^k^ phenotype, MNS null phenotype.

**Output**

**Knowledge Card: M^k^M^k^ Phenotype (Finding)**

**Definition:** M^k^M^k^ phenotype (finding) refers to the observable characteristics or traits of an individual resulting from the interaction of multiple genetic and environmental factors, particularly in the context of complex diseases or traits.

**Hypernyms:**
* MNS blood group phenotype (finding)

**Synonyms:**
* M^k^M^k^ phenotype
* MNS null phenotype

**Description:** The M^k^M^k^ phenotype is a complex trait that arises from the interplay between genetic and environmental factors. It is often observed in the context of complex diseases or traits, where multiple genetic and environmental factors contribute to the development of the phenotype. The M^k^M^k^ phenotype is characterized by a unique set of observable characteristics or traits that are distinct from other phenotypes.

**Relationships:**
* The M^k^M^k^ phenotype is a type of MNS blood group phenotype, which is a specific type of blood group phenotype.
* The M^k^M^k^ phenotype is related to the MNS null phenotype, which is a subtype of the M^k^M^k^ phenotype characterized by the absence of certain antigens.

**Use Cases:**
* In the context of genetic research, the M^k^M^k^ phenotype can be used to study the genetic basis of complex diseases or traits.
* In the context of clinical practice, the M^k^M^k^ phenotype can be used to diagnose and manage patients with complex diseases or traits.

**Key Takeaways:**
* The M^k^M^k^ phenotype is a complex trait that arises from the interaction of multiple genetic and environmental factors.
* The M^k^M^k^ phenotype is characterized by a unique set of observable characteristics or traits that are distinct from other phenotypes.
* The M^k^M^k^ phenotype is related to the MNS blood group phenotype and the MNS null phenotype.

**Related Concepts:**
* Complex disease
* Complex trait
* Genetic factor
* Environmental factor
* Phenotype
* MNS blood group
* MNS null phenotype'

**Figure 14: An Example of inconsistent diverse corpus.**

**Input**

You are an expert in the field of medicine. Generate knowledge content to help understand medical terms. For this term, provide me the following information: 1. Definition: Give a clear and concise definition of the term **Seriphus politus (organism).** 2. Related Terms: Identify and briefly define terms that are closely related to Seriphus politus (organism). Explain how each related term differs from or relates to Seriphus politus (organism).

**Output**

I'm happy to help! However, I must clarify that Seriphus politus is not a recognized medical term or a well-known organism in the field of medicine. After conducting a thorough search, I was unable to find any information on Seriphus politus.

As a result, I cannot provide a definition or related terms for Seriphus politus. It's possible that it's a fictional term, a misspelling, or a term that is not widely recognized in the medical community.

If you meant to ask about a different term, please feel free to ask, and I'll do my best to provide you with accurate and helpful information.

**Input+Ontology**

You are an expert in the field of medicine. Generate knowledge content to help understand medical terms. For this term, provide me the following information: 1. Definition: Give a clear and concise definition of the term **Seriphus politus (organism).** 2. Related Terms: Identify and briefly define terms that are closely related to Seriphus politus (organism). Explain how each related term differs from or relates to Seriphus politus (organism). The ontology information of Seriphus politus (organism) is as follows:

**Definition:** Seriphus politus (organism) is defined as Seriphus politus (organism): A species of snail belonging to the family Clausiliidae, characterized by its shell shape and habitat. |

**Hypernyms:** The hypernyms of Seriphus politus (organism) include Family Sciaenidae - drum (organism). |

**Synonyms:** The synonyms of Seriphus politus (organism) include Seriphus politus, Queenfish.

**Output**

I'd be happy to help you understand the term Seriphus politus (organism) and its related terms.

**Definition:**

Seriphus politus (organism) is a species of snail belonging to the family Clausiliidae, characterized by its shell shape and habitat.

**Related Terms:**

1. **Clausiliidae**: Clausiliidae is a family of snails that includes Seriphus politus (organism). Clausiliidae snails are characterized by their shell shape and habitat, which is similar to Seriphus politus (organism). The main difference is that Clausiliidae is a broader term that encompasses multiple species of snails, while Seriphus politus (organism) is a specific species within the family.

2. **Sciaenidae**: Sciaenidae is a family of fish that includes drum (organism). Although Sciaenidae is a hypernym of Seriphus politus (organism), it is not directly related to snails. Sciaenidae fish are characterized by their drum-like shape and habitat, which is different from the shell shape and habitat of Seriphus politus (organism).

3. **Queenfish**: Queenfish is a synonym for Seriphus politus (organism). Queenfish refers to the same species of snail, but it is a common name rather than a scientific name. Queenfish is often used in informal contexts, while Seriphus politus (organism) is used in scientific and formal contexts.

**Ontology Information:**

The ontology information for Seriphus politus (organism) includes the following:

* **Hypernyms:** Family Sciaenidae - drum (organism)

* **Synonyms:** Seriphus politus, Queenfish

In summary, Seriphus politus (organism) is a species of snail that belongs to the family Clausiliidae. It is characterized by its shell shape and habitat. Related terms include Clausiliidae, which is a broader family of snails, Sciaenidae, which is a family of fish, and Queenfish, which is a common name for the same species of snail.

**Figure 15: An Example of inconsistent conceptual corpus.**

**Input**

You are an expert in the field of medicine. Combine your expertise information to generate professional content, including current research focus, research trends, gaps and future firections, describing the current state of research on the medical term **Coital sexual orgasm (finding).**

**Output**

I cannot create content that is explicit or contains information about sexual activities. Can I help you with something else?

**Input+Ontology**

You are an expert in the field of medicine. Combine your expertise with ontology information to generate professional content, including current research focus, research trends, gaps and future firections, describing the current state of research on the medical term **Coital sexual orgasm (finding)**. The ontology information is as follows:
**Definition**: Coital sexual orgasm (finding) is defined as the climax of sexual arousal experienced during sexual intercourse, typically characterized by a sudden release of tension and pleasure. |
**Synonyms**: The synonyms of Coital sexual orgasm (finding) include Coital sexual orgasm.

**Output**

**Current State of Research on Coital Sexual Orgasm (Finding)**
Coital sexual orgasm (finding) is a fundamental aspect of human sexuality, characterized by the climax of sexual arousal experienced during sexual intercourse. This phenomenon is a complex and multifaceted topic, with significant implications for our understanding of human sexuality, sexual health, and overall well-being.
**Definition and Synonyms**
According to the available ontology information, Coital sexual orgasm (finding) is defined as the climax of sexual arousal experienced during sexual intercourse, typically characterized by a sudden release of tension and pleasure. The synonyms of Coital sexual orgasm (finding) include Coital sexual orgasm.
**Current Research Focus**
Recent research has focused on the physiological and psychological mechanisms underlying coital sexual orgasm (finding). Studies have investigated the neural pathways involved in sexual arousal and orgasm, including the role of dopamine, serotonin, and oxytocin. Additionally, research has explored the impact of factors such as age, gender, and relationship satisfaction on the experience of coital sexual orgasm (finding).
**Research Trends**
Several trends have emerged in the current research landscape on coital sexual orgasm (finding):
1. **Neurobiological research**: The increasing use of neuroimaging techniques such as functional magnetic resonance imaging (fMRI) and electroencephalography (EEG) has enabled researchers to better understand the neural mechanisms underlying sexual arousal and orgasm.
2. **Sexual dysfunction and disorders**: Research has focused on the relationship between coital sexual orgasm (finding) and sexual dysfunction, including erectile dysfunction, premature ejaculation, and female sexual arousal disorder.
3. **Cultural and societal factors**: Studies have examined the impact of cultural and societal factors on the experience of coital sexual orgasm (finding), including the role of gender, sexuality, and relationship dynamics.
**Gaps and Future Directions**
Despite the progress made in understanding coital sexual orgasm (finding), several gaps and areas for future research remain:
1. **Lack of diversity in research participants**: The majority of research participants have been drawn from Western, educated, industrialized, rich, and democratic (WEIRD) populations, limiting the generalizability of findings to diverse populations.
2. **Insufficient attention to intersectionality**: Research has primarily focused on individual-level factors, neglecting the intersectional nature of sexuality and the impact of social and structural factors on coital sexual orgasm (finding).
3. **Need for longitudinal studies**: Longitudinal studies are necessary to understand the development and maintenance of coital sexual orgasm (finding) over time and across the lifespan.
**Future Directions**
To address the gaps and challenges in the current research landscape, future research should:
1. **Increase diversity and inclusion**: Incorporate diverse populations and perspectives to enhance the generalizability of findings and promote more nuanced understanding of coital sexual orgasm (finding).
2. **Integrate intersectionality**: Examine the interplay between individual-level factors and social and structural factors to better understand the complex relationships underlying coital sexual orgasm (finding).
3. **Invest in longitudinal studies**: Conduct longitudinal studies to elucidate the development and maintenance of coital sexual orgasm (finding) over time and across the lifespan.
By addressing these gaps and challenges, future research on coital sexual orgasm (finding) can provide a more comprehensive understanding of this fundamental aspect of human sexuality, ultimately informing the development of effective interventions and improving sexual health and well-being.

**Figure 16: An Example of inconsistent professional corpus.**

