# OpenReview forum: "OntoTune: Ontology-Driven Self-training for Aligning Large Language Models"
_ACM.org/TheWebConf/2025/Conference — WWW 2025 Oral_

### Official Review · Reviewer_Mu7N · 2024-11-27

**Novelty:** 6
**Technical Quality:** 6

**Review:**

In this work, the authors propose an ontology-driven self-training framework for aligning large language models. The proposed pipeline aims to inject domain-specific knowledge (medical ontology) into LLMs without requiring training on a large domain-specific corpus. The paper is well-written, and the experimental results are comprehensive and solid. The pipeline is well-described and I do not have too many technical issues. Some of my major comments are:
1.	While this work presents itself as a general fine-tuning framework, the experiments are primarily conducted using a medical ontology and corpus. The lack of evaluations on other domains may limit the generalizability of the work. Are well-defined ontologies readily available in other domains to support similar applications?
2.	Do the embedding-based baselines include BERT models pretrained on medical corpora?

**Questions:**

Please refer to the comments above.

**Reviewer Confidence:**

3: The reviewer is confident but not certain that the evaluation is correct

**Scope:**

4: The work is relevant to the Web and to the track, and is of broad interest to the community

---

### Official Review · Reviewer_diXY · 2024-11-28

**Novelty:** 5
**Technical Quality:** 5

**Review:**

The paper presents OntoTune, an ontology-driven self-training framework designed to align large language models (LLMs) with domain-specific knowledge by leveraging hierarchical conceptual structures like ontologies. Instead of relying on large-scale domain-specific corpora, OntoTune reorganizes LLM knowledge using in-context learning to identify gaps and guide further training. The framework is evaluated in the medical domain using SNOMED CT, demonstrating state-of-the-art performance in hypernym discovery and medical domain QA tasks while preserving the general and safety-related knowledge of the seed models.

strengthens:
OntoTune introduces a self-training paradigm that systematically integrates ontology knowledge into LLMs without requiring extensive domain-specific corpora. This is a meaningful step toward more efficient and cost-effective domain adaptation. The framework achieves superior results over existing methods like TaxoLLaMA and domain-specific LLMs trained on large corpora. The performance gains in hypernym discovery tasks and medical QA highlight the practical advantages of OntoTune. By focusing on small-scale, structured knowledge sources (ontologies), OntoTune addresses the issue of fragmented domain knowledge in LLMs. The results show improved alignment and generalization, even outperforming more resource-intensive approaches. The approach demonstrates strong adaptability across different datasets, tasks, and even seed models, showcasing its versatility and robustness.

**Questions:**

1: While the paper emphasizes in-context learning for ontology alignment, it would be valuable to discuss other possible ways to incorporate in-context learning. For example, could multi-modal in-context vectors [1,2] or embeddings be explored to enrich the integration of ontological knowledge?

[1] Liu, Rui, et al. "In-context learning for zero-shot medical report generation." Proceedings of the 32nd ACM International Conference on Multimedia. 2024.
[2] Liu, Sheng, et al. "In-context vectors: Making in context learning more effective and controllable through latent space steering." arXiv preprint arXiv:2311.06668 (2023).

2: The performance of OntoTune’s three variants (SFT, DPO, and SFT+DPO) shows inconsistencies, particularly across tasks. Could the authors provide more analysis on why some methods work better in specific cases, and why there is no clear overall trend? For instance, results fluctuate across MedMCQA and USMLE tasks.

3: Ontologies often encompass multi-modal data (e.g., images in medical diagnosis). How would OntoTune handle the integration of multi-modal ontological knowledge? Could this be a future direction for improving the versatility of the framework?

4: While OntoTune outperforms TaxoLLaMA in aligning LLMs with domain-specific ontologies, are there any cases where direct ontology injection might be preferable? For example, in tasks that require exhaustive domain-specific reasoning, could OntoTune’s reliance on inconsistencies lead to gaps?

**Reviewer Confidence:**

3: The reviewer is confident but not certain that the evaluation is correct

**Scope:**

3: The work is somewhat relevant to the Web and to the track, and is of narrow interest to a sub-community

---

### Official Review · Reviewer_vzvQ · 2024-12-02

**Novelty:** 5
**Technical Quality:** 6

**Review:**

A well written, structured and documented paper with a robust evaluation and interesting results. A nice contribution to a vibrant research area worth a discussion.

Some typos that should be mended:
p1, l23 – compared
p1, l111 – these mind maps
p2, l203 – developed
p3, l326 – extent
p5, l567 – observe(d)
p6, l668 – a large scale corpus v large scale corpora v at large scale corpora
p6, l669 – although

**Questions:**

Can you briefly comment the generalizability of your approach and its applicability to other knowledge domains, i.e. w.r.t. examples mentioned in your introductory statement? Are there any plans to engage in this direction?

p6, l 663: Can you be more specific on “small-scale” and “high quality” ontologies? Is there any reference you can provide or metrics that can be applied to formalize / operational the claim laid out in this passage? Where would you position the SNOMED ontology in a hypothetical spectrum of small and large scale ontologies? Did you test how ontology size and quality impacts the results? Do you have an opinion on this?

**Reviewer Confidence:**

2: The reviewer is willing to defend the evaluation, but it is likely that the reviewer did not understand parts of the paper

**Scope:**

4: The work is relevant to the Web and to the track, and is of broad interest to the community

---

### Official Review · Reviewer_s6bC · 2024-12-02

**Novelty:** 5
**Technical Quality:** 6

**Review:**

The authors highlight the limitations of domain-specific LLM development, which relies on injecting large-scale domain corpora, and propose OntoTune, a fine-tuning method based on ontology. OntoTune outperformed baselines on benchmarks such as hypernym discovery and domain QA. Interestingly, it also improves performance in some general knowledge tasks.

**Pros**
+ Proposes an effective method for incrementally training LLMs with ontological knowledge.
+ Achieves high accuracy compared to existing methods on multiple benchmarks.
+ Reports extensive experiments and some unexpected but interesting findings.

**Cons**
- Little relevance to web technologies.
- The technical originality is limited as a submission to the Semantics and Knowledge track.
- Insufficient evaluation of whether the LLM aligns with the ontological knowledge.

**Quality**
The paper sets three research questions and conducts extensive experiments to address them. It also provides some analysis of the model, including robustness to seed models and training parameters.

**Clarity**
The paper is well-organized. However, it occasionally omits explanations of existing methods and datasets, requiring readers to check numerous references, which is not reader-friendly.

**Originality**
The technical originality is limited as a submission to the **Semantics and Knowledge** track. Template-based instruction is not novel. The originality of the proposed method lies in the fine-tuning process based on Inconsistency Text Selection to reorganize LLM's domain knowledge.

**Significance of this work**
The proposed method is effective in reorganizing LLM's domain knowledge using ontological knowledge. Its simplicity offers the potential for diverse developments in the future.

**Detailed Comments**

Regarding RQ1, evaluating whether the model aligns with ontological knowledge is challenging, and more than hypernym discovery alone is required.

Generally, domain ontologies include complex semantic relationships such as class and property restrictions in addition to hypernyms and synonyms. Hypernym is relatively simple domain knowledge that seed models can often answer, which raises the possibility that ontological knowledge might not be as critical. Additionally, there is no guarantee that the knowledge addressed is unlearned by the LLM—it might merely reinforce the frequency of knowledge already learned.

Experiments focusing on knowledge exclusively found in the ontology, rather than general knowledge common to both documents and ontology, might yield more interesting results. However, I understand that suitable benchmark datasets for such evaluations are unavailable.

For instance, tasks involving complex ontology matching [1] might offer a higher-level evaluation of whether the model aligns with ontological knowledge.

[1] Survey on complex ontology matching, http://doi.org/10.3233/SW-190366

**Questions:**

I'm curious whether SFT, DPO, and SFT+DPO are truly effective.
Did the authors compare OntoTune with approaches based solely on in-context learning? Specifically, was Instruction+Ontology in Figure 2 included as a baseline for comparison?

**Reviewer Confidence:**

3: The reviewer is confident but not certain that the evaluation is correct

**Scope:**

3: The work is somewhat relevant to the Web and to the track, and is of narrow interest to a sub-community